# Neural Network Modeling and Dynamic Analysis of Different Types of Engine Mounts for Internal Combustion Engines

**DOI:** 10.3390/s22051821

**Published:** 2022-02-25

**Authors:** Jessimon Ferreira, Bianca Marin, Giane G. Lenzi, Calequela J. T. Manuel, José M. Balthazar, Wagner B. Lenz, Adriano Kossoski, Angelo M. Tusset

**Affiliations:** Federal University of Technology—Paraná, Ponta Grossa 84017-220, Brazil; jessimon@alunos.utfpr.edu.br (J.F.); bianca.marin@continental.com (B.M.); gianeg@utfpr.edu.br (G.G.L.); calequelamanuel@alunos.utfpr.edu.br (C.J.T.M.); balthazar@utfpr.edu.br (J.M.B.); wagner.1991@alunos.utfpr.edu.br (W.B.L.); kossoski@alunos.utfpr.edu.br (A.K.)

**Keywords:** passive vibration isolators, vehicle dynamics, quarter-car model, passive control, artificial neural networks

## Abstract

This paper presents the results of studies on reducing the amount of vibrations in different frequency ranges generated by a combustion engine through the use of different types of engine mounts. Three different types of engine supports are experimentally and numerically analyzed, namely an elastomeric engine mount, an elastomeric engine mount with a hydraulic component and standard decoupling, and an elastomeric engine mount with a hydraulic component and a modified decoupler—with this engineering design being a novelty in the literature. Experimental tests that considered different excitation frequencies were performed for the three types of engine mounts. Experimental data for stiffness and damping were used to obtain nonlinear mathematical models of the two systems with hydraulic components through the use of an Artificial Neural Network (ANN). For the results, all of the mathematical models presented coefficients of determination, R^2^, greater than 0.985 for both stiffness and damping, showing an excellent fit for the nonlinear experimental data. Numerical results using a quarter-car suspension model showed a large reduction in vibration amplitudes for the first vibration model when using the hydraulic systems, with values ranging between 48.58% and 66.47%, depending on the tests. The modified system presented smaller amplitudes and smoother behavior when compared to the standard hydraulic model.

## 1. Introduction

According to [1], mechanical vibrations have always been a major concern in modern engineering systems such as vehicles, aerospace systems and other types of applications. The suppression of unwanted vibrations is also essential for more complex systems because they can lead to loss of performance, reduced lifetime and premature failures that can lead to safety issues [2,3]. For motorized propulsion systems, the study of vibrations is essential, since as well as causing mechanical failure of the project, vibration can be responsible for passenger discomfort, directly affecting how people respond to the system.

The internal combustion engine is the most common source of propulsion for vehicles, transforming the chemical energy present in different types of fuels into mechanical energy used for movement. Due to the engine’s operating principles, such as internal combustion and the great number of moving parts, a wide range of frequency vibrations arise inside it, which are transferred to all parts of the vehicle. Therefore, with the growing demand for comfort combined with performance and energy efficiency, studies on and solutions for reducing vibration and noise levels are essential to the current modern automotive industry [4,5,6].

A vehicle’s engine mounting system is usually composed of several supports connected to the vehicle’s structure [7]. Engine mounts have the function of supporting the weight of the engine as well as isolating the rest of the vehicle—including the passengers—from vibration and noise generated by the engine. Unwanted vibrations come from two different sources: those of high amplitude and low frequency come from the road and those of low amplitude and high frequency are generated by the engine [8,9,10].

To mitigate the vibrations that arise inside the motor, the vast majority of modern cars use passive engine mounts, such as a conventional rubber support (elastomeric) or a hydraulic support. Between these two types, the elastomeric support is the most common, due especially to its small dimensions and lower cost when compared to other solutions [11]. However, due to its material characteristics, it may perform unsuitably at low resonant frequencies [7]. Hydraulic engine mounts provide alternative solutions to elastomeric mounts due to their ability to modify their performance according to different frequencies [12]. However, just like rubber supports, hydraulic supports are also unable to act in all possible frequencies that may arise from normal engine behavior [8,13,14].

With regards to passenger comfort, the search for engineering solutions that can improve performance in current automotive systems by reducing noise and vibration transmitted from the engine to the chassis has attracted the attention of engineers and researchers [4,5,15]. 

According to [13], these engine mounts can be modeled as a cubic spring. However, the load at the end of the frequency spectrum has stronger nonlinearities and the model needs corrections. Another point that also significantly increases the modeling complexity is the presence of fluid. In [15], the graphic modeling of a viscoelastic absorber is presented, considering parameters obtained with the use of a parameter identification system and data obtained through finite elements (FE). Numerical and experimental results have shown the influence of frequency on the nonlinearities of viscoelastic absorbers. 

This paper presents an experimental and numerical investigation of three different models of engine mounts, in order to contribute to the research conducted by automotive engineers in projects that use computer simulations and mathematical models. The three cases developed in this work are an elastomeric engine mount, an elastomeric engine mount with a hydraulic component and standard decoupling (similar to the one studied by [15]), and an elastomeric engine mount with a hydraulic component and modified decoupler. The modification in the decoupler, which is new in the literature, consists of the inclusion of a 3 mm diameter hole in the decoupler, creating a direct passage between the compensation chamber and the main chamber, for additional fluid passage. 

Figure 1a shows a generic model of an elastomeric engine mount with a hydraulic component similar to those used in the experimental tests of this work. Figure 1b shows the elastomeric engine mount with a hydraulic component and modified decoupler proposed for this paper.

In Figure 1, (1) and (2) are the pins used to secure the elastomeric engine mount with the hydraulic component between the engine and the vehicle chassis; (3) is a rubber structure capable of supporting the weight of the engine; and (4) and (5) are the upper and lower chambers, respectively. During engine movement and vibrations, the internal fluid transits between these two chambers. The decoupler, a path with low resistance to the fluid movement, is labeled as (6). The decoupler absorbs part of the fluid, and the rest of the fluid is directed along the so-called inertia track (7), which presents a greater resistance to the displacement of the fluid. The decoupler is free to move between the top plate (8) and the bottom plate (9). The diaphragm (10) is responsible for allowing the lower chamber to expand or contract. The modified decoupler proposed for the elastomeric absorber support with a hydraulic component is labeled (11).

In order to evaluate the vibration reduction capacity between the engine and the chassis for different frequency ranges of engine vibration, we present the experimental analysis of three different types of elastomeric engine mount and propose the mathematical models for the coefficients of stiffness and damping for the three cases studied. 

An Artificial Neural Network (ANN) with backpropagation was used to obtain the dynamic models of the engine mounts with hydraulic components due to its greater complexities and nonlinearities. The necessary training data were obtained experimentally. A backpropagation method was selected because it allows fast convergence with the data obtained experimentally, therefore, better processing time and lower costs in developing the model [16]. Using the experimental data when designing the neural network allows for a dynamic model that represents the nonlinear forces acting on the vibration isolating system. This enables us to obtain a model that better reflects reality and can be used to expand studies and to predict real behaviors through numerical simulations [16]. To verify the effectiveness of each tested support in reducing vibration levels, we present the dynamic analysis of a quarter-car model with an engine coupling and engine mount using the mathematical models obtained from the studied supports. The quarter-car suspension model, even though it is simple in conception, is widely used and validated in several studies as a testing and comparison platform for new solutions and systems [17,18]. Thus, the use of this type of suspension is justified in this work, which aims to test three different types of engine mount in a standardized suspension system. 

The remainder of this article is organized as follows: Section 2 presents the experimental results of the dynamic stiffness coefficient and damping for an elastomeric engine mount, an elastomeric engine mount with a hydraulic component with standard decoupler, and an elastomeric engine mount with a hydraulic component with a modified decoupler. Section 3 presents the mathematical models obtained as well as the structure of the neural network used for this purpose. Section 4 presents the dynamic analysis of an engine-coupled quarter-car system using the studied engine mount supports. The final section presents the conclusions.

## 2. Materials and Methods

The equipment used to obtain the experimental results was an elastomer testing machine model 831.50 with features for high-frequency testing, manufactured by MTS^®^. The data acquisition system used was the FlexTest GT, which works with MTS software 793. Experimental tests were performed considering a harmonic excitation signal given by the following: (1)y=Asin(ωt)
where *A* is the excitation amplitude and ω is the frequency excitation given in rad/s.

Each sample was tested in 3 different amplitudes: ±0.01 mm, ±0.2 mm, and ±2.0 mm. The frequency range for the tests was from 1 to 40 Hz. These amplitudes and frequencies are related to the majority of the engine rotations and thus are of the greatest interest for control and suppression [15,19].

Figure 2 presents the experimental results of the dynamic stiffness of the supports for different excitation frequencies.

As can be seen in Figure 2a (case 1), the dynamic stiffness of the elastomeric engine mount is lower than that observed in the elastomeric engine mount with a hydraulic component being presented in Figure 2b,c. This extra attenuation comes from the fluid present inside the engine mount chamber of the engine mount with a hydraulic component. One can also observe that the orifice added to the decoupler (case 3, Figure 2c) caused a reduction in the dynamic stiffness when compared to the results obtained for the traditional decoupler (case 2, Figure 2b), with this being especially visible at lower excitation amplitudes (±0.1 mm and ±0.2 mm). 

Figure 3 shows the experimental results of the damping for all engine mounts.

As can be seen in Figure 3a, the damping capability presented by the elastomeric engine mount is also lower than that observed for the elastomeric engine mount with a hydraulic component, thus showing the importance of the presence of fluid inside the elastomeric support chamber. Analyzing the results presented in Figure 3c, it is observed that the hole added to the decoupler caused the damping coefficient to decrease compared to the support with the standard decoupler (Figure 3b), demonstrating greater impact at smaller excitation amplitudes (±0.1 mm and ±0.2 mm). The reduction in damping is consistent with what was expected, as the orifice decreased the resistance to flow, thus allowing for rapid fluid movement. 

## 3. Mathematical Modeling 

In order for a numerical simulation to correspond to a real system with a certain level of confidence, a dynamic model of the system as complete as possible is also necessary. However, the more complete the dynamic model, the more expensive it becomes to obtain this model due to the high mathematical cost, especially when working with complex dynamic systems. In many mathematical designs and models, the nonlinearities of a system are generally disregarded to reduce the mathematical difficulty. However, in some cases, the nonlinear effect is an intrinsic behavior of the system and cannot be removed from the model. Thus, a very important step in the process is the choice of techniques that will be used to obtain the model.

When analyzing the stiffness coefficient for case 1 (Figure 2a), an increment behavior can be observed, which can be represented by a power function in the form $K_{case_{1}}=\beta_{1}\omega^{\beta_{2}}$, where $\beta_{1}$ and $\beta_{2}$ are constants. Using the least squares method, one can obtain the following: (2)$$K_{case_{1}}=2648.5\omega^{-0.807}$$
where ω is the frequency (rad/s). 

For the damping coefficient for case 1 (Figure 3a), a logarithmic decrement can be observed. Thus, the function $C_{case_{1}}=\sigma_{1}\ln(\omega )+\sigma_{2}$ can be used, where $\sigma_{1}$ and $\sigma_{2}$ are the constants to be determined. Applying the least squares method, we have the following: (3)$$C_{case_{1}}=3899.2\ln(\omega )+128820$$

However, when observing the cases referring to systems with hydraulic components given by Figure 2b,c and Figure 3b,c, a strong nonlinear behavior is observed. Thus, machine learning techniques can be used to obtain more accurate models in a shorter period of time, using the process experimental data to fit the numerical model to real-life behavior [20]. In this paper, the use of an artificial neural network with backpropagation is proposed. The network structure used to obtain the mathematical models is of the multilayer type, composed of an input layer with five neurons, two hidden layers with five neurons in each of these layers and an output layer with one neuron, as can be observed in Figure 4.

Five neurons were used in the input layer of the ANN. The activation function used was the sigmoidal hyperbolic tangent, as it represents a continuous range between −1 and 1, being able to adequately process the available experimental data. The output layer has only one neuron, which has a linear function as activation. Thus, the models that are approximate to using the neural network are expected to have the following general function: (4)$$f(\omega )=\sum_{i}^{n}\frac{\delta_{1}}{1+e^{\delta_{2}\omega +\delta_{3}}}+\delta_{4}$$
where the parameters $\delta_{1}$, $\delta_{2}$, $\delta_{3}$ and $\delta_{4}$ come from the neural network. Thus, all of the damping and stiffness have the same structure. 

In this study, the use of the proposed artificial neural network and the described activation functions is justified by the ability of this type of system to capture the complex nonlinear relationship in the experimental data, incorporating all of the influential parameters. The advantages of using this type of network are also observed in [20].

For the damping coefficient and stiffness in case 2 (elastomeric engine mount with a hydraulic component with standard decoupling), the mathematical models obtained through the employed artificial neural network are represented in Equations (5) and (6), respectively:(5)$$\begin{array}{l} C_{case_{2}}=\frac{6159.5}{e^{\left(0.0345\omega -3.779\right)}+1}-\frac{76497}{e^{\left(-0.1704\omega +12.405\right)}+1}-\frac{22645}{e^{\left(-0.2855\omega +18.624\right)}+1}\\ -\frac{306233}{e^{\left(0.099\omega -6.886\right)}+1}+\frac{198922}{e^{\left(0.0957\omega -7.2034\right)}+1}+103299 \end{array}$$
(6)$$\begin{array}{l} K_{case_{2}}=\frac{627544}{e^{\left(0.096\omega -9.253\right)}+1}-\frac{4656.6}{e^{\left(-0.037\omega +6.572\right)}+1}+\frac{6176.2}{e^{\left(0.1179\omega -16.193\right)}+1}\\ +\frac{34128}{e^{\left(0.1137\omega -4.6879\right)}+1}-\frac{139422}{e^{\left(0.19854\omega -12.87\right)}+1}+203533 \end{array}$$

For case 3 (elastomeric engine mount with a hydraulic component and modified decoupler), the mathematical models that represent the damping and stiffness coefficient, obtained through the ANN, are represented in Equations (7) and (8), respectively: (7)$$\begin{array}{l} C_{case_{3}}=\frac{11851}{e^{\left(0.0358\omega -4.1636\right)}+1}+\frac{201199}{e^{\left(-0.081635\omega +5.971\right)}+1}-\frac{132577}{e^{\left(-0.0808\omega +6.4803\right)}+1}\\ +\frac{53025}{e^{\left(0.12459\omega -9.2458\right)}+1}+\frac{1112.5}{e^{\left(0.05769\omega -12.387\right)}+1}-63925 \end{array}$$
(8)$$\begin{array}{l} K_{case_{2}}=\frac{78.804}{e^{\left(3.4588\omega -791.37\right)}+1}-\frac{111811}{e^{\left(0.1135\omega -8.1273\right)}+1}-\frac{31434}{e^{\left(-0.10577\omega +5.0149\right)}+1}\\ -\frac{7670.6}{e^{\left(-0.03409\omega +6.4162\right)}+1}-\frac{2553.7}{e^{\left(-0.11511\omega +15.396\right)}+1}+2424 \end{array}$$

Obtaining mathematical models of the engine mount stiffness and damping coefficients allows these models to be applied to numerical simulations that can determine the transmission of vibrations from the engine to the chassis or even to the vehicle’s passengers.

### Validation of the Proposed Models

In Figure 5, it is possible to observe the numerical results of the proposed models and the experimental data when considering the stiffness of the supports for case 1 (elastomeric engine mount), case 2 (elastomeric engine mount with a hydraulic component with standard decoupling) and case 3 (elastomeric engine mount with a hydraulic component and modified decoupler).

As can be seen, the curves obtained for the stiffness of the mathematical models presented a good fit to the experimental data, and for case 1, the coefficient of determination, R^2^, was 0.9872. For cases 2 and 3, the obtained adjustment was even better, where for both, the coefficient of determination, R^2^, was 0.9999. This concludes that, although the studied models are complex, the use of only five neurons in the input layer was enough to obtain a good correlation between the experimental data and the models obtained by the proposed ANN. The addition of more neurons or intermediate layers would only increase the computational cost and not bring a significant improvement in the results.

Figure 6 shows the numerical results of the proposed models and the experimental data for the damping of the supports for case 1 (elastomeric engine mount), case 2 (elastomeric engine mount with a hydraulic component) and case 3 (elastomeric engine mount with a hydraulic component and modified decoupler), respectively.

The mathematical models for damping obtained by the ANN showed a good fit to the experimental data. This fact can be validated from the coefficient of determination. For the elastomeric absorber support (case 1), it was R^2^ = 0.9864, and for cases 2 and 3, it was R^2^ = 0.9999. Thus, the neural networks proposed for this work were validated and represent the system behavior in a satisfactory way.

## 4. Numerical Simulations for a Quarter-Car Model Subject to Engine Vibration 

Automotive suspensions can be classified into three different groups: passive, semi-active and active. This classification is given according to the suspension force on the system [21]. It is called passive when the suspension force comes only from the springs and dampers. In this type of system, all parameters are fixed and depend on the type of material, design and project. In semi-active suspensions, the damping component can vary its magnitudes within a range through the insertion of external energy [21,22]. Generally, data from sensors and a control system are used to vary the actuation force within certain limits. In the active system, sensors and controllers are also used; however, the system must always remain active because there are no base values [21]. These systems are generally more complex and costly to maintain. 

The purpose of the simulations in this section is to study how the different types of engine mount act when in the presence of vibrations coming from the engine and how these vibrations are passed on to the vehicle’s suspension system. A passive suspension model was chosen. Since the objective is to analyze only the engine mounts, the passive suspension model is the best for comparison tests because it isolates the results from the engine mounts without influencing the results with external systems. For the suspension, a quarter-car model is used to analyze the influence of each type of engine mount on the vibrations transmitted to the car chassis, with the car engine as the source of vibration. The choice of this model is due to its high ability in simulating vertical displacement movements of vehicles. It can provide sufficient information about the behavior of a vehicle suspension system, being widely used in various types of research. The determination of a quarter-car model consists of isolating a quarter of the vehicle’s suspension. For vehicles with evenly distributed weight, the results are very close to a full-car model [23,24,25,26,27].

Quarter-vehicle models generally have only two degrees of freedom: the vertical displacement of the suspended mass and the unsprung mass. For the case where the motor is included, the system will have three degrees of freedom, as shown in Figure 7.

Here, $m_{s}$ represents the mass of the body, $m_{u}$ is the wheel axle mass, $m_{e}$ is the engine mass, $b_{m}$ is the engine mount damping, $k_{m}$ is the stiffness of engine mount, $b_{s}$ represents the passive damper of a conventional suspension structure, $k_{s}$ is the spring set, $k_{t}$ represents the tire as a bundle of springs, $x_{r}$ represents vertical movements of the tire, $x_{u}$ represents vertical movements of the wheel, $x_{s}$ represents vertical movements of the body and $x_{m}$ represents vertical movements of the engine [28,29,30].

The position of the chassis axis is considered as the coordinate reference point $x_{c}$, where $k_{s}\left(x_{s}-x_{u}\right)$ is the stiffness force $k_{s}$, $b_{s}({\dot{x}}_{s}-{\dot{x}}_{u})$ is the damping force $b_{s}$, $k_{t}(x_{r}-x_{u})$ is the strength due to the stiffness of the tires $k_{t}$, ${\ddot{x}}_{s}$ is the acceleration of the body mass $m_{s}$, $k_{m}(x_{e}-x_{s})$ represents the stiffness force of the support, $b_{m}({\dot{x}}_{e}-{\dot{x}}_{s})$ represents the damping force of the support, ${\ddot{x}}_{s}$ is the acceleration of the chassis mass and $m_{s}$ and ${\ddot{x}}_{u}$ are the accelerations of the axle and wheel mass $m_{u}$, respectively. The displacements and speed of the suspension elements are given by the relative movement between the bodies and the movements of the engine.

Applying Newton’s Second Law, ∑F=ma, in each mass element separately, the Force system for the quarter-car model in Figure 7 can be represented by the following [28,29,30]: (9)$$\begin{array}{l} m_{s}{\ddot{x}}_{s}=-b_{s}^{l}({\dot{x}}_{s}-{\dot{x}}_{u})+b_{s}^{y}\left|{\dot{x}}_{s}-{\dot{x}}_{u}\right|-b_{s}^{nl}\sqrt{\left|{\dot{x}}_{s}-{\dot{x}}_{u}\right|}\mathrm{sgn}({\dot{x}}_{s}-{\dot{x}}_{u})\\ -k_{s}^{l}\left(x_{s}-x_{u}\right)-k_{s}^{nl}{\left(x_{s}-x_{u}\right)}^{3}+b_{m}({\dot{x}}_{e}-{\dot{x}}_{s})+k_{m}(x_{e}-x_{s})\\ m_{u}{\ddot{x}}_{u}=b_{s}^{l}({\dot{x}}_{s}-{\dot{x}}_{u})-b_{s}^{y}\left|{\dot{x}}_{s}-{\dot{x}}_{u}\right|+b_{s}^{nl}\sqrt{\left|{\dot{x}}_{s}-{\dot{x}}_{u}\right|}\mathrm{sgn}({\dot{x}}_{s}-{\dot{x}}_{u})\\ +k_{s}^{l}\left(x_{s}-x_{u}\right)+k_{s}^{nl}{\left(x_{s}-x_{u}\right)}^{3}-k_{t}(x_{r}) \end{array}$$
where $x_{e}=\alpha \sin\left(\omega t\right)$ and ${\dot{x}}_{e}=\alpha \omega \cos\left(\omega t\right)$. The coefficient $k_{s}^{l}$ represents the linear actuation range and the nonlinear coefficient, $k_{s}^{nl}$ represents the nonlinear characteristic of the spring, the coefficient $b_{s}^{l}$ affects the damper force in a linearly way while the coefficient $b_{s}^{nl}$ acts in a nonlinear way on the damper, and the coefficient $b_{s}^{y}$ represents the characteristics of the asymmetric behavior of the damper.

For the numerical simulations, the internal combustion engine is considered a source of excitation. All of the physical parameters used in the simulations are presented in Table 1 [28,29,30].

When using the internal combustion engine as a constant excitation source and knowing its displacement, it is possible to analyze the efficiency of the proposed mathematical models for the supports. The maximum displacements of the chassis for different amplitudes are represented in Figure 8.

The results in Figure 8a show a reduction in displacement of 48.94% for the elastomeric engine mount with a hydraulic component with standard decoupling (case 2) and 48.58% for the elastomeric engine mount with a hydraulic component and modified decoupler (case 3) for the first natural frequency mode when compared to the elastomeric support. For the second natural frequency mode, the reduction was 15.79% for the elastomeric engine mount with a hydraulic component with standard decoupling (case 2) in relation to the elastomeric engine mount with a hydraulic component and modified decoupler (case 3). Additionally, for Figure 8b, the results show a reduction in displacement of 48.74% for the elastomeric engine mount with a hydraulic component with standard decoupling (case 2), and 48.60% for the elastomeric engine mount with a hydraulic component and modified decoupler (case 3) for the first mode of natural frequency. For the second natural frequency mode, the reduction was 15.86% for the elastomeric engine mount with a hydraulic component with standard decoupling (case 2) in relation to the engine mount support with a hydraulic component and modified decoupler (case 3). Finally, the results presented by Figure 8c show a reduction in displacement of 65.91% for the elastomeric engine mount with a hydraulic component with standard decoupling (case 2) and 66.47% for the elastomeric engine mount with a hydraulic component and modified decoupler (case 3) for the first mode of natural frequency. For the second natural frequency mode, the reduction was 14.09% for the elastomeric engine mount with a hydraulic component with standard decoupling (case 2) in relation to the elastomeric engine mount with a hydraulic component and modified decoupler (case 3).

At high excitation frequencies, it is noted that the chassis displacement amplitude for the case of the elastomeric engine mount (case 1) is smaller than for the cases of the elastomeric engine mount with a hydraulic component with standard decoupling (case 2) and for the elastomeric engine mount with a hydraulic component and modified decoupler (case 3). This is due to the fact that practically all vibrations coming from the engine are transmitted to the chassis. Therefore, for higher excitation frequencies, the elastomeric engine mount with a hydraulic component with standard decoupling and the elastomeric engine mount with a hydraulic component and modified decoupler present better vibration attenuation capabilities when compared to the elastomeric mount.

## 5. Conclusions

The results obtained in this paper demonstrate that the damping provided by the elastomeric engine mount is inferior to the elastomeric engine mount with a hydraulic component when comparing vibration amplitudes of the first mode of vibration. Through the results obtained by numerical simulations, it is observed that, because the dynamic stiffness of the elastomeric engine mount increases as the excitation frequency increases, the transmission of vibration from the engine to the chassis is greater than that observed for the elastomeric engine mount with a hydraulic component, demonstrating the advantage of using this type of system. It was also possible to observe that the addition of a small hole in the decoupler, connecting the compensation chamber to the main chamber, has an effect on damping, especially at small excitation amplitudes. This effect is a result of the increase in the flow area, which reduces the system’s resistance to fluid flow between the chambers. It is observed that the modified system presented better results for frequencies in the region of 58 rad/s when compared to the standard models. These results contribute to the study of automotive engineering, demonstrating that small modifications in a decoupler can be used to create an engine mount that meets pre-established requirements. Additionally, the nonlinear mathematical models obtained can be used as a starting point for this type of technology development. 

The designed neural network proved to be capable of obtaining excellent mathematical models that represent the nonlinear behavior of the absorbers. All of the models showed an excellent correlation with the experimental data, where none presented a coefficient of determination lower than 0.985. These results contribute to previous research, providing a mathematical model with a higher correlation than that obtained by the graphical modeling in [15].

## Figures and Tables

**Figure 1 sensors-22-01821-f001:**
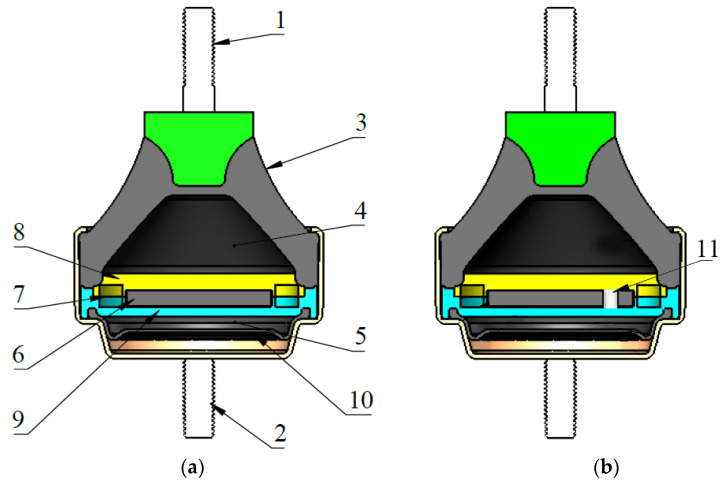
Cross section of the engine mount support with hydraulic component: (**a**) standard model; (**b**) model with the modified decoupler.

**Figure 2 sensors-22-01821-f002:**
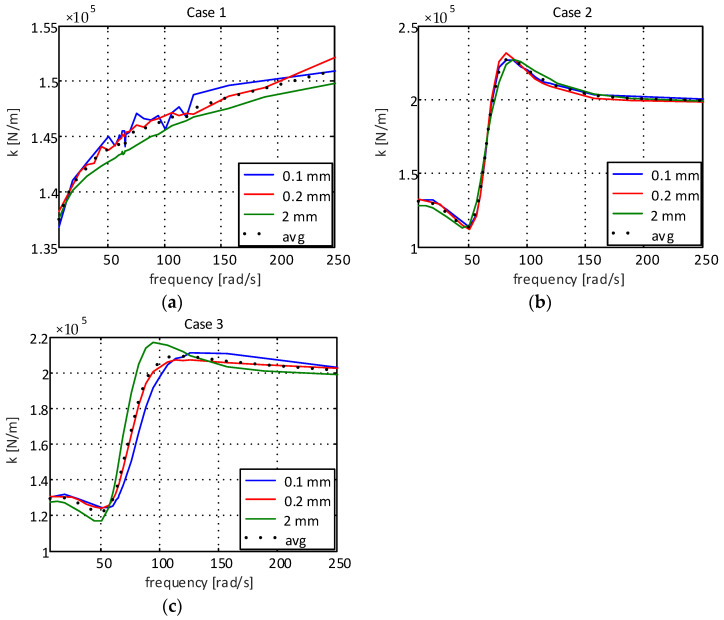
Dynamic stiffness coefficient of the supports: (**a**) elastomeric engine mount; (**b**) elastomeric engine mount with hydraulic component and standard decoupling; (**c**) elastomeric engine mount with hydraulic component and modified decoupler.

**Figure 3 sensors-22-01821-f003:**
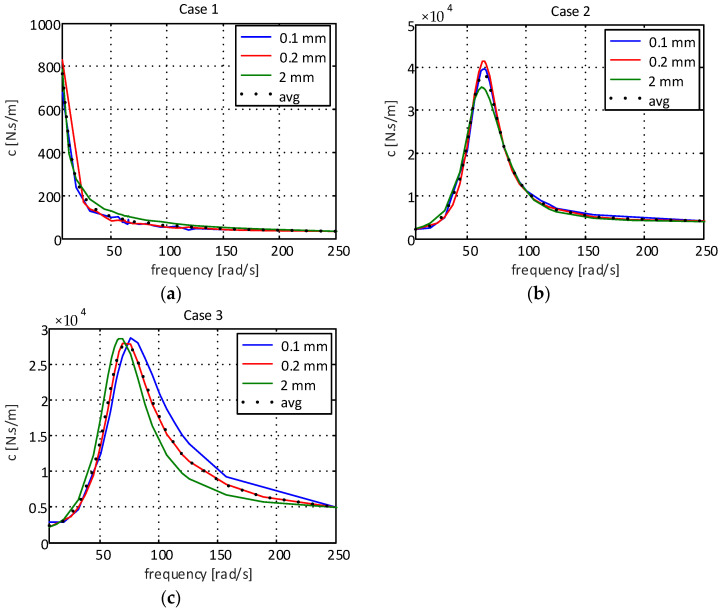
Damping coefficient of the supports: (**a**) elastomeric engine mount; (**b**) elastomeric engine mount with hydraulic component and standard decoupling; (**c**) elastomeric engine mount with hydraulic component and modified decoupler.

**Figure 4 sensors-22-01821-f004:**
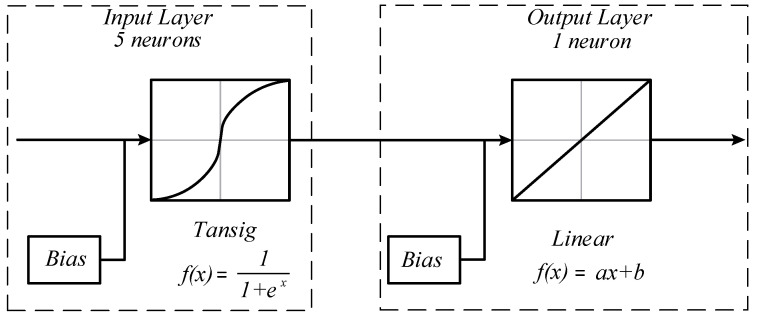
Artificial neural network structure.

**Figure 5 sensors-22-01821-f005:**
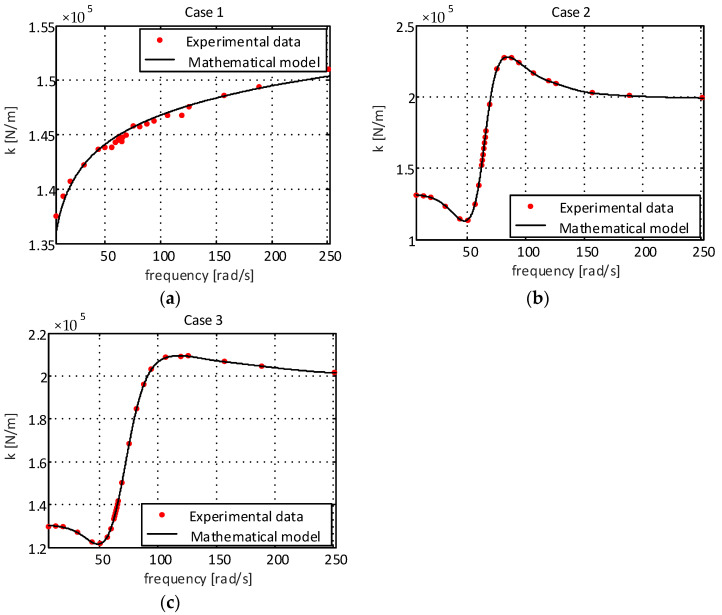
Stiffness coefficient of the supports: (**a**) elastomeric engine mount; (**b**) elastomeric engine mount with hydraulic component with standard decoupling; (**c**) elastomeric engine mount with hydraulic component and modified decoupler.

**Figure 6 sensors-22-01821-f006:**
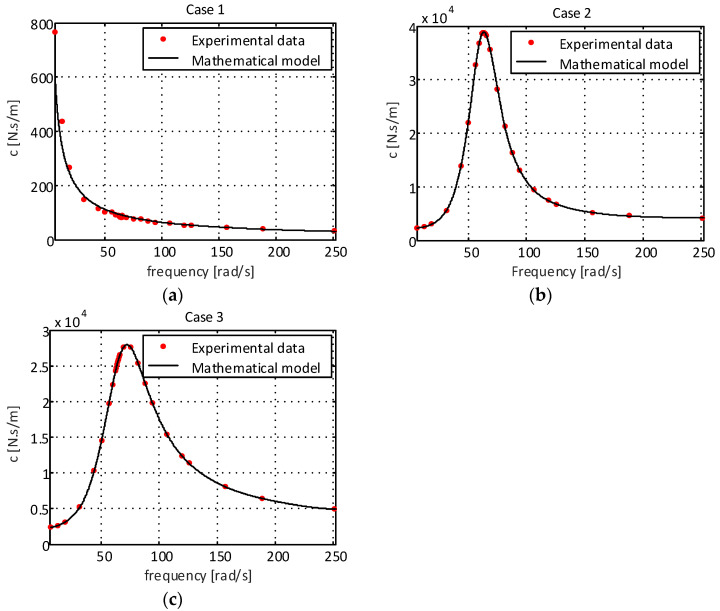
Damping coefficients of the supports: (**a**) elastomeric engine mount; (**b**) elastomeric engine mount with hydraulic component with standard decoupling; (**c**) elastomeric engine mount with hydraulic component and modified decoupler.

**Figure 7 sensors-22-01821-f007:**
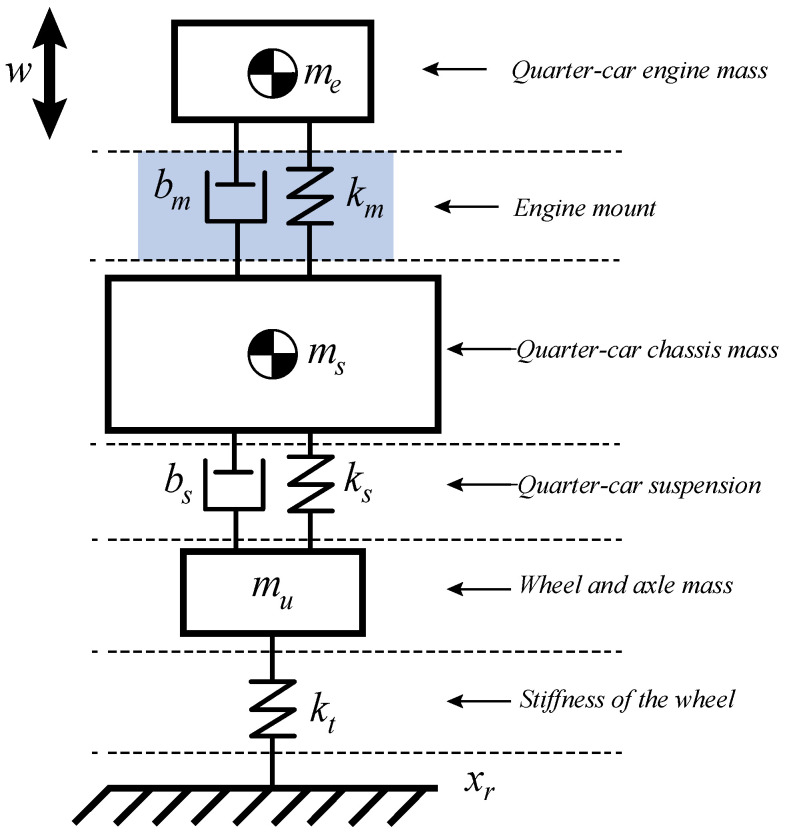
Quarter-car model with the engine and supports.

**Figure 8 sensors-22-01821-f008:**
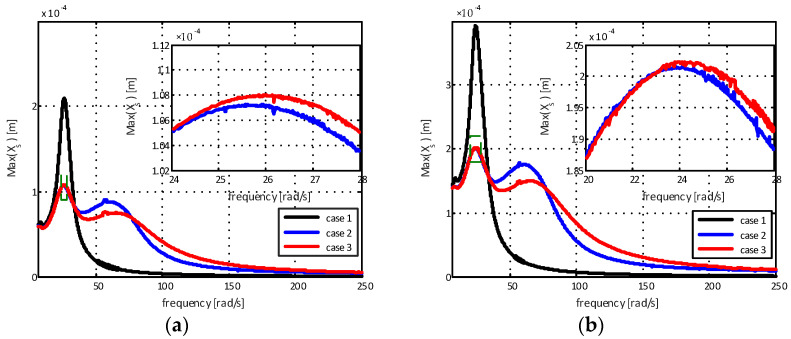
Maximum chassis displacement for different amplitudes: (**a**) $\alpha =10^{-4}$ m; (**b**) $\alpha =2\times 10^{-4}$ m; (**c**) $\alpha =2\times 10^{-3}$ m.

**Table 1 sensors-22-01821-t001:** Numerical parameters.

Parameter	Value (Unity)
$m_{s}$	375 (kg)
$m_{u}$	59 (kg)
$b_{s}^{l}$	700 (Ns/m)
$b_{s}^{nl}$	200 (Ns/m)
α	10^−4^; 2 × 10^−4^; 2 × 10^−3^ (m)
$b_{s}^{y}$	400 (Ns/m)
$k_{s}^{l}$	235 × 10^2^ (N/m)
$k_{s}^{nl}$	235 × 10^4^ (N/m)
$k_{t}$	190 × 10^3^ (N/m)
ω	[0:250] (rad/s)

## Data Availability

Not applicable.

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
