# Peer review of "Neural Network Modeling and Dynamic Analysis of Different Types of Engine Mounts for Internal Combustion Engines"

_sensors, 2022, doi:10.3390/s22051821_

Round 1

Reviewer 1 Report

[In the title]

It is suggested that the main method can be reflected in the title. Particularly, the artificial intelligence method or Neural network can be reflected in the title.

[In the abstract]

Some statistical results can be given to demonstrate the effectiveness of the present model.

[In the introduction]

It is suggested to indicate the motivation of suppressing mechanical vibration in some complex coupling systems, such as [1-2], as the vibration may cause the degradation of the performance and lead to some safety issues.

[1] Modeling and a cross-coupling compensation control methodology of a large range 3-DOF micropositioner with low parasitic motions." Mechanism and Machine Theory 162 (2021): 104334.

[2] A spatial coupling model to study dynamic performance of pantograph-catenary with vehicle-track excitation, Mech. Syst. Signal Process. 151 (2021) 107336.

After the literature review in Line 60, the potential solution should be discussed before presenting the hydraulic support in Figure 1.

[In section 2]

Please give one sentence to explain why these three amplitudes are selected in the experimental test.

[In section 3]

The input and output layers should be included in Figure 4 to facilitate the reader to follow.

[In section 4]

If the variable b represents the nonlinear damper, why it is given by a constant value in Table 1?

Author Response

We would like to thank the reviewers for spending their time in reading, reviewing, and commenting on our manuscript. Those comments are all valuable and very helpful for revising and improving our manuscript to a better scientific level.
We have studied the raised comments carefully and made corrections, which we hope that meet your requirements. Please, consider the reviewers' comments in black, the authors' answers in blue and the changes made to the paper in red (Attached).

Reviewer 2 Report

This paper presents and analyzes the capability to reduce different frequency range vibrations generated by an internal combustion engine using different type passive vibration isolators. Three different types of engine supports are experimentally analyzed, namely: an elastomeric support, a hydraulic support with a standard decoupler and a hydraulic support with a standard decoupler and considering an extra hole in the structure. The experimental data of stiffness and damping are used to obtain mathematical models for the three isolation systems through the use of an artificial neural network with a backpropagation mechanism. Numerical simulations of a quarter-car suspension system are carried out in order to evaluate the damping and restoration characteristics of the three vibration insulators studied through comparative results.

The topic is interesting, and the manuscript is comprehensive. However, the novelty of the work is difficult to appreciate. The authors should describe what is the contribution and originality of this paper compared with most recent research papers. References are grouped and associated to very general topics. Then, the authors must show some criticism on some of the existing works and prove they bring something new.

Authors' conclusion that 'the engine coupling shows that the use of passive isolators significantly reduces vibrations and that hydraulic supports present better results when compared to simple rubber insulators. I consider it unsustainable, considering the simple model used to represent a quarter-car. Furthermore, this calls into question all the conclusions of the paper.

Author Response

(The authors gave the same response as above.)

Reviewer 3 Report

This paper presents a study in the context of passive vibration control to reduce unwanted vibrations generated by an internal combustion engine. An experimental analysis for three different types of engine supports is described. Mathematical models for the coefficient of stiffness and damping for the three cases are proposed. I have the following comments:

1) The innovation of this proposal with respect to other similar contributions should be clearly described and justified.

2) This study considers the passive vibration control like an alternative to reduce different frequency range vibrations generated by an internal combustion engine. Three different types of engine supports are analyzed for passive vibration control. It is recommended to describe how this proposal could be extended or combined with semi-active and active vibration control implementations.

3) Section of Introduction should be complemented with recent contributions published in journals. Certainly, passive vibration control has been widely implemented to reduce unwanted vibrations in vehicle suspensions. However, there are active and semi-active vibration control techniques for vehicle suspension systems. Authors should describe the main difference of this proposal with respect to the recent contributions: doi.org/10.3390/act10030047, doi.org/10.3390/act9030077.

4) Mathematical modeling of the supports is carried out using artificial neural networks with backpropagation. Section 3 should be improved to clearly describe the derivations of equations for damping and stiffness.

5) An explanation why artificial neural networks were used for the mathematical models. It is suggested to describe if other mathematical structures for damping coefficient and stiffness could be used, for instance, polynomial models.

6) There are several architectures artificial neural networks. Explain how the artificial neural network structure shown in Fig. 4 was selected for mathematical modelling purposes.

Author Response

(The authors gave the same response as above.)

Round 2

Reviewer 1 Report

All my comments have been well addressed. 

Reviewer 2 Report

The authors have answered all issues. The additions are sufficient, and the manuscript has been improved. Therefore, I think the paper is worth to be published.

Reviewer 3 Report

I have no further suggestions. This paper may be accepted in present form now.